# NatCat: Weakly Supervised Text Classification with Naturally Annotated Resources

**Zewei Chu**                                                     ZEWEICHU@GMAIL.COM
*University of Chicago, Chicago, IL 60637, USA*

**Karl Stratos**                                                 KARLSTRATOS@GMAIL.COM
*Rutgers University, Piscataway, NJ 08854, USA*

**Kevin Gimpel**                                                   KGIMPEL@TTIC.EDU
*Toyota Technological Institute at Chicago, Chicago, IL 60637, USA*

## Abstract

We describe NATCAT, a large-scale resource for text classification constructed from three data sources: Wikipedia, Stack Exchange, and Reddit. NATCAT consists of document-category pairs derived from manual curation that occurs naturally within online communities. To demonstrate its usefulness, we build general purpose text classifiers by training on NATCAT and evaluate them on a suite of 11 text classification tasks (CATEVAL), reporting large improvements compared to prior work. We benchmark different modeling choices and resource combinations and show how tasks benefit from particular NATCAT data sources.[1]

## 1. Introduction

Websites with community contributed content contain ample knowledge for natural language processing. In this paper, we seek to improve text classification by leveraging this knowledge in the form of texts paired with natural category annotations. In particular, we present NATCAT, a large-scale resource constructed automatically from three data sources: Wikipedia, Stack Exchange,[2] and Reddit.[3] With rich knowledge about text and categories, we show that NATCAT is a useful resource for text classification.

To demonstrate the usefulness of NATCAT, we use it to train models that compute a score for any document-category pair. As a result, we get weakly supervised (i.e., "dataless"; Chang et al., 2008) text classifiers that can be used off-the-shelf to produce interpretable and relevant topics for any document. They can also be effortlessly ported to a specific topic classification task by computing the score for only the labels in the task. Table 1 illustrates the use of a NATCAT-trained model on a document from AGNEWS.

To evaluate, we propose CATEVAL, a standardized benchmark for evaluating weakly supervised text classification with a choice of datasets, label descriptions for each dataset, and baseline results. CATEVAL comprises a diverse choice of 11 text classification tasks including both topic and sentiment labels, and contains both single and multi-label classification tasks. We show that NATCAT is a valuable resource for weakly supervised text classification and study the impact of data domain and pretrained model choice on tasks in CATEVAL. We analyze the gap between weakly supervised and supervised models and show that the

---

1. NATCAT is available at https://github.com/ZeweiChu/NatCat
2. https://stackexchange.com/
3. https://www.reddit.com/

| Text | Israeli ambassador calls peace conference idea 'counterproductive'. A broad international peace conference that has reportedly been suggested by Egypt could be "counterproductive" and shouldn't be discussed until after ... |
|---|---|
| **AGNews** | ***international***, *sports, science technology, business* |
| **NatCat (Wiki.)** | *invasions, diplomats, peace, diplomacy, environmentalism, Egypt, patriotism ...* |

Table 1: An example from AGNews, showing the text, the provided AGNews categories (true class in bold), and predicted categories from a text classifier trained on the Wikipedia version of NatCat (in decreasing order by score). The NatCat-trained model can be used to score any document-category pair, so it can be applied to the AGNews task by scoring each category and choosing the one with the highest score. The AGNews row shows the category ordering from the NatCat classifier.

mistakes of our models are reasonable and humanlike, suggesting that the construction process of NatCat is a promising approach to mining data for building text classifiers.

## 2. The NatCat Resource

In this section, we describe the creation of the NatCat resource. NatCat is constructed from three different data sources with natural category annotation: Wikipedia, Stack Exchange, and Reddit. Therefore, NatCat naturally contains a wide range of world knowledge that is useful for topical text classification. For each data source, we describe ways of constructing document-category pairs in which the document can be labeled with the category.

**Wikipedia.** Wikipedia documents are annotated with categories by the community contributors. The categories of each Wikipedia document can be found at the bottom of the page. We obtained Wikipedia documents from Wikimedia Downloads. Wikipedia page-to-category mappings were generated from Wiki SQL dumps using the "categorylinks" and "page" tables. We removed hidden categories by SQL filtering, which are typically maintenance and tracking categories that are unrelated to the document content. We also removed disambiguation categories. After filtering, there are 5.75M documents with at least one category, and a total of 1.19M unique categories. We preprocessed the Wikipedia articles by removing irrelevant information such as the external links at the end of each article. We then removed Wikipedia documents with fewer than 100 non-stopwords.

Some category names are lengthy and specific, e.g., "Properties of religious function on the National Register of Historic Places in the United States Virgin Islands". These categories are unlikely to be as useful for end users or downstream applications as shorter and more common categories. Therefore, we consider multiple ways of augmenting the given categories with additional categories.

The first way is to use a heuristic method of breaking long category names into shorter ones. We first use stopwords as separators and keep each part of the non-stopword word sequence as a category name. For each category name of a document, we also run a named entity recognizer [Honnibal and Montani, 2017] to find all named entities in that category

|                        | Wikipedia | Stack Exchange | Reddit    |
|------------------------|-----------|----------------|-----------|
| # categories           | 1,730,447 | 156            | 3,000     |
| # documents            | 2,800,000 | 2,138,022      | 7,393,847 |
| avg. # cats. per doc.  | 86.9      | 1              | 1         |
| mode # cats. per doc.  | 46        | 1              | 1         |
| avg. # words per doc.  | 117.9     | 58.6           | 11.4      |

Table 2: Statistics of training sets sampled from NatCat for each of its three data sources.

name, and add them to the category set of the document. This way we expand the existing category names from Wikipedia. For the example category above, this procedure yields the following categories: "religious function", "the national register of historic places", "properties", "historic places", "the united states virgin islands", "properties of religious function on the national register of historic places in the united states virgin islands", "united states virgin islands", and "national register".

Our second method of expansion is based on the fact that Wikipedia categories can have parent categories and therefore form a hierarchical structure. When expanding the category set by adding its ancestors, there is a trade-off between specificity/relevance and generality/utility of category names. Using only the categories provided for the article yields a small set of high-precision, specific categories. Adding categories that are one or two edges away in the graph increases the total number of training pairs and targets more general/common categories, but some of them will be less relevant to the article. In NatCat, we include all categories of documents that are up to two edges away.

**Stack Exchange.** Stack Exchange is a question answering platform where users post and answer questions as a community. Questions on Stack Exchange fall into 308 subareas, each area having its own site. We create document-category pairs by pairing question titles or descriptions with their corresponding subareas. Question titles, descriptions, and subareas are available from Chu et al. [2020]. Many Stack Exchange subareas have their own corresponding "meta" sites. A meta site is meant to discuss the website itself regarding its policy, community, and bugs, etc. When creating this dataset, we merge the subareas with their corresponding meta areas. This gives us over 2 million documents with 156 categories.

**Reddit.** Inspired by Puri and Catanzaro [2019], we construct a category classification dataset from Reddit. In our dataset, we propose to classify Reddit post titles to their corresponding subreddit names. We use the OpenWebText[4] toolkit to get Reddit posts with more than 3 karma and their subreddit names. We keep only the top 3000 most frequent subreddits as they better capture the common categories that we are interested in.[5] This gives us over 7 million documents with 3000 categories.

---

4. https://github.com/jcpeterson/openwebtext

5. Subreddit names are generally more noisy than categories in the other two data sources, and many are abbreviations or shorthand that are meaningful only to particular groups. Wikipedia category names are more formal and understandable by most people. Hence we decide to only keep the most common categories from Reddit. Some example subreddit names (with ranking by frequency in parentheses):

| dataset | # test docs. | # labels | # sents./doc. | # words/doc. | # words/sent. |
|---|---|---|---|---|---|
| AGNews | 7,600 | 4 | 1.3 | 48.8 | 36.8 |
| DBpedia | 70k | 14 | 2.4 | 58.7 | 24.4 |
| Yahoo | 60k | 10 | 5.7 | 115.8 | 20.3 |
| 20 News Groups | 7,532 | 20 | 15.9 | 375.4 | |
| Emotion | 16k | 10 | 1.6 | 19.5 | 12.4 |
| SST-2 | 1,821 | 2 | 1.0 | 19.2 | 19.1 |
| Yelp-2 | 38k | 2 | 8.4 | 155.1 | 18.4 |
| Amazon-2 | 400k | 2 | 4.9 | 95.7 | 19.5 |
| NYTimes | 10k | 100 | 30.0 | 688.3 | 22.9 |
| Comment | 1,287 | 28 | 1.3 | 13.8 | 10.5 |
| Situation | 3,525 | 12 | 1.8 | 44.0 | 24.7 |

Table 3: Statistics of CatEval datasets.

Constructed by these three data sources, NatCat covers a wide range of topics and world knowledge. Table 2 summarizes statistics of training sets we sampled from NatCat. Note that all documents from Stack Exchange and Reddit have only one associated category, while a document from Wikipedia may have multiple categories describing it.

## 3. CatEval Tasks

To evaluate NatCat, we will use it to build general purpose text classifiers and test them on a variety of text classification tasks. In this section, we introduce CatEval, which comprises a diverse choice of 11 text classification tasks including both topic-related and sentiment-related labels, and contains both single and multi-label classification tasks. For single label topic classification, we have AGNews,[6] DBpedia [Lehmann et al., 2015], Yahoo [Zhang et al., 2015], and 20 News Groups [Lang, 1995].

For sentiment classification, we use Emotion [Klinger et al., 2018], SST-2 [Socher et al., 2013], Yelp-2, and Amazon-2 [Zhang et al., 2015]. Emotion is a fine-grained sentiment classification tasks with labels expressing various emotions, while the other three are binary sentiment classification tasks differentiating positive and negative sentiments.

As for multi-label topical classification, we have NYTimes [Sandhaus, 2008], Comment,[7] and Situation [Mayhew et al., 2019] datasets. The NYTimes categories have hierarchical structure, but we merely use the category names from the lowest level. We removed newspaper-specific categories that are not topical in nature.[8] Of the remaining 2295 categories, we only use the 100 most frequent categories in our experiments, and randomly sample 1 million documents for the training set, 10k for a dev set, and 10k as a test set.[9]

---

archeage (3001), Deutsche_Queers (10000), HartfordWhalers (20000), pyladies (30000), Religitards (40000), TheArnoldFanClub (50000), HomeSeer (60000), 001 (68627).

6. https://www.di.unipi.it/~gulli/AG_corpus_of_news_articles.html

7. https://dataturks.com/projects/zhiqiyubupt/comment

8. *opinion*, *paid death notices*, *front page*, and *op-ed*

9. Training and dev sets are only used for the supervised baselines.

Table 3 summarizes the key statistics of each dataset, including the average number of sentences, average number of words, and average sentence length. They cover a broad range of text classification tasks and can serve as a benchmark for text classifiers.

**Choice of label names.** For a given document classification task, we need to specify the name of each label. As in prior work [Chang et al., 2008, Song and Roth, 2014], we manually choose words corresponding to labels in the downstream tasks. Our models and the baselines use the same label names which are provided in the appendix. In recent work, we showed empirically the variance of dataless classifiers due to label name variations, and proposed unsupervised methods to reduce this variance for particular datasets [Chu et al., 2021].

As NatCat is a large textual resource with ample categories, almost all labels in the CatEval datasets appear in NatCat except for some conjunction phrases, such as "written work", "manufacturing operations and logistics", and "home and garden". However, there is no guarantee that the labels in NatCat have the same definition as the labels in the downstream tasks, and in fact we find such divergences to be causes of error, including when measuring human performance on the text classification tasks. Weakly supervised methods (and humans) are more susceptible to semantic imprecision in label names than supervised methods.

**Why is this weakly supervised?** The goal of weakly supervised text classification is to classify documents into any task-specific categories that are not necessarily seen during training. We build models with NatCat that are capable of scoring any candidate label for any document. To evaluate, we take the test sets of standard document classification tasks, use the model to score each label from the set of possible labels for that task, and return the label with the highest score. The reason we describe our method as "weakly supervised" is because it does not require annotated training data with the same labeling schema and from the same distribution as the test set, but rather uses freely-available, naturally-annotated document/category pairs as a training resource.

**Existing applications.** Even though the category distributions of NatCat do not exactly match the target distribution in a downstream task, generalization is possible if they are sufficiently similar and cover a diverse set of knowledge. This setting is referred to using several different terms, including dataless classification [Chang et al., 2008], transfer learning [Pan and Yang, 2009], distant supervision [Mintz et al., 2009], and weakly-supervised learning [Zhou, 2017].

## 4. Experiments

### 4.1 Models and Training

We fine-tune BERT [Devlin et al., 2019] and RoBERTa [Liu et al., 2019] on NatCat. In our experiments, we use BERT-base-uncased (110M parameters) and RoBERTa-base (110M parameters). We formulate NatCat training as a binary classification task to predict whether a category correctly describes a document. Binary cross entropy loss is used for training. For each document-category pair, we randomly sample 7 negative categories for training. As documents from Wikipedia have multiple positive categories, we randomly choose one positive category. To form the input for both BERT and RoBERTa, we concatenate the

category with the document: "[CLS] category [SEP] document [SEP]". In our experiments, we truncate the document to ensure the category-document pair is within 128 tokens.

For BERT and RoBERTa models shown in Table 4, we train models on the whole NatCat dataset and also separately the data from each single data source (Wikipedia, Stack Exchange, or Reddit) for comparison. For each single source, we train the model for one epoch on 100k instances. When training on all three sources, we train on 300k instances. The learning rate is set to be 0.00002, and we perform learning rate warmup for 10% of the training steps and then linearly decay the learning rate. As BERT and RoBERTa models are known to suffer from randomness among different fine-tuning runs, we perform each single experiment 5 times with different random seeds and report the median of such five runs. We also do supervised training on Emotion, NYTimes, Comment, and Situation with the RoBERTa model. We follow the same training procedure as we train on NatCat to solve a document-category binary classification task.

We compare NatCat-trained models to Explicit Semantic Analysis (ESA; Gabrilovich and Markovitch, 2007) as our primary baseline.[10] ESA was shown effective for dataless classification by Chang et al. [2008], and we mostly follow their procedure. However, instead of setting a threshold on the number of concepts, we use all concepts as we find this improves ESA's performance. In preliminary experiments, we tried other unsupervised text representation learning approaches, e.g., encode the document and category using pretrained models or pretrained word embeddings (BERT, ELMo, GloVe, etc.), then use cosine similarity as the scoring function for a document-category pair. We also experimented with the pretrained GPT2 model and fine-tuning it on NatCat [Radford et al., 2019]. However, we found these methods do not perform as well as weakly supervised approaches such as ESA and our approach, so we do not report the results of such methods in this paper.

## 4.2 Evaluation

We report classification accuracy for all single label classification tasks, including topical and sentiment tasks. For multi-label classification tasks, we use label ranking average precision (LRAP), a multi-label generalization of the mean reciprocal rank. For each example $i = 1 \ldots N$, let $\mathcal{Y}^{(i)} \subseteq \{1 \ldots m\} = \mathcal{Y}$ denote the set of gold labels and $f^{(i)} \in \mathbb{R}^m$ denote the model label scores. LRAP is defined as

$$\text{LRAP}\left(\left\{\mathcal{Y}^{(i)}, f^{(i)}\right\}_{i=1}^N\right) = \frac{1}{N} \sum_{i=1}^N \frac{1}{|\mathcal{Y}^{(i)}|} \sum_{y_{\text{gold}} \in \mathcal{Y}^{(i)}} \frac{\left|\left\{z_{\text{gold}} \in \mathcal{Y}^{(i)} : f^{(i)}_{z_{\text{gold}}} \geq f^{(i)}_{y_{\text{gold}}}\right\}\right|}{\left|\left\{y \in \mathcal{Y} : f^{(i)}_y \geq f^{(i)}_{y_{\text{gold}}}\right\}\right|}$$

which achieves the highest value of 1 iff all gold labels are ranked at the top. To directly compare with Yin et al. [2019] in multi label classification tasks, we also use label-wise weighted F1 score in some cases.

## 4.3 Primary Results

Table 4 summarizes the experimental results of BERT and RoBERTa models trained on NatCat and evaluated on CatEval. RoBERTa trained on NatCat performs the best on

---

10. Using code from github.com/CogComp/cogcomp-nlp/tree/master/dataless-classifier

| | Topical (Acc.) | | | | Sentiment (Acc.) | | | | | Multi label (LRAP) | | | | |
|---|---|---|---|---|---|---|---|---|---|---|---|---|---|---|---|
| | ag | dbp | yah | 20n | avg | emo | sst | yel | amz | avg | nyt | com | sit | avg | all |
| NatCat Trained Models (BERT, RoBERTa, and ensembles) | | | | | | | | | | | | | | | |
| B | **75.6** | 82.8 | 54.9 | 39.3 | 63.3 | 16.1 | 62.7 | 70.4 | 63.6 | 53.8 | 49.6 | **22.6** | 50.5 | 41.0 | 53.3 |
| e | 75.4 | 83.0 | 55.2 | **41.7** | **63.8** | 16.6 | 65.7 | 67.6 | 68.4 | 54.6 | **50.8** | **22.6** | 50.8 | **41.4** | 54.3 |
| R | 68.8 | 81.9 | 57.8 | 36.8 | 61.3 | 21.2 | 65.0 | 67.3 | 66.8 | 55.8 | 47.7 | 21.5 | 52.3 | 40.5 | 53.4 |
| e | 68.4 | **85.0** | **58.5** | 37.6 | 62.4 | **22.3** | **68.7** | **75.2** | 72.4 | **59.7** | 49.0 | 22.1 | 52.6 | 41.2 | **55.6** |
| Other weakly supervised models (ESA, GPT2) | | | | | | | | | | | | | | | |
| E | 71.2 | 62.5 | 29.7 | 25.1 | 47.1 | 9.5 | 52.1 | 51.1 | 51.9 | 41.2 | 10.9 | 22.5 | **55.6** | 29.7 | 40.2 |
| G | 65.5 | 44.8 | 49.5 | - | - | - | 62.5 | 74.7 | **80.2** | - | - | - | - | - | - |
| Fully supervised and human performances | | | | | | | | | | | | | | | |
| S | 92.4 | 98.7 | 71.2 | 85.5 | 87.0 | 34.5 | 97.1 | 95.6 | 95.1 | 80.6 | 72.5 | 64.7 | 75.2 | 70.8 | 76.4 |
| H | 83.8 | 88.2 | 75.0 | - | - | - | - | - | - | - | - | - | - | - | - |

Table 4: Results of BERT (B) and RoBERTa (R) trained on NatCat and evaluated on CatEval. "e" are ensembles over NatCat-trained models with 5 random seeds. We compare with the dataless classifier ESA (E) [Chang et al., 2008] which we ran on our datasets and GPT2 (G) used as a text classifier as reported by Puri and Catanzaro [2019]. We also compare to supervised results (S). All is the average over all 11 tasks. The highest weakly-supervised result per column is in boldface.

average across tasks, but there are some differences between BERT and RoBERTa. BERT is better on AGNews and NYTimes, both of which are in the newswire domain, as well as 20NG, which also involves some news- or technical-related material. RoBERTa is better on Yahoo as well as better on average in the emotion, binary sentiment, and situation tasks. This may be due to RoBERTa's greater diversity of training data (web text) compared to BERT's use of Wikipedia and books.

To provide perspective on the difficulty of the weakly supervised setting, we obtained annotations from 3 human annotators involved in this research project on 60 instances from AGNews, 50 from DBpedia, and 100 from Yahoo. We showed annotators instances and the set of class labels and asked them to choose a single category using their own interpretation and judgment without the ability to look at any training examples. Average accuracies from three annotators on these tasks are reported in Table 4.

We also compare with results from supervised methods. The results of AGNews, DBpedia, Yahoo, Yelp-2 and Amazon-2 are from Zhang et al. [2015]. The SST-2 result is from Wang et al. [2019]. The 20 News Groups result is from Pappagari et al. [2019]. NYTimes, Situation, Comment and Emotion results are fine-tuned RoBERTa models.

In some tasks (AGNews and DBpedia), supervised models outperform human annotators. We believe this is caused by semantic drift between human interpretation and the actual meaning of the labels as determined in the dataset. Supervised models are capable of learning such nuance from the training data, while an annotator without training is not capable of classifying documents in that way. Weakly supervised models are like human

| | Yahoo [Yin et al., 2019] | | | | Emotion | | | | Situation | | | |
| | v0 | | v1 | | v0 | | v1 | | v0 | | v1 | |
| | seen | unseen | seen | unseen | seen | unseen | seen | unseen | seen | unseen | seen | unseen |
|---|---|---|---|---|---|---|---|---|---|---|---|---|
| Half seen setting | | | | | | | | | | | | |
| BERT | 73.2 | 12.9 | 80.9 | 9.9 | 32.8 | 17.8 | 34.8 | 20.3 | 73.1 | 50.4 | 63.9 | 41.6 |
| + NatCat | 73.6 | 16.5 | 80.1 | 12.2 | 33.4 | 17.6 | 34.6 | 19.4 | 72.8 | 48.6 | 61.9 | 41.3 |
| Yin et al. | *72.6* | *44.3* | *80.6* | *34.9* | 35.6 | 17.5 | 37.1 | 14.2 | 72.4 | 48.4 | 63.8 | 42.9 |
| NatCat-trained fully unseen setting | | | | | | | | | | | | |
| BERT | 57.3 | 52.5 | 52.5 | 57.3 | 18.2 | 14.6 | 14.6 | 18.2 | 38.4 | 35.7 | 35.7 | 38.4 |
| RoBERTa | 63.5 | 52.1 | 52.1 | 63.5 | 21.4 | 11.4 | 11.4 | 21.4 | 44.0 | 30.8 | 30.8 | 44.0 |

Table 5: Experiments in half seen (where dataset instances for some labels are seen during training and some are not) and fully unseen settings (our weakly supervised setting that does not use any training instances from the target datasets).

annotators in that they are only capable of classifying documents with the general knowledge they have learned (in this case from large scale naturally-annotated data).

Yin et al. [2019] build weakly supervised text classifiers from Wikipedia. Comparing with their reported F1 results in a zero-shot setting (Yahoo: 52.1; Emotion: 21.2; Situation: 27.7), NatCat-trained BERT models yield better results in topic classification tasks (Yahoo: 54.9; Situation: 37.1), though not as strong on others (Emotion: 16.1), proving that NatCat is a valuable resource of topical knowledge.

### 4.4 Half Seen Settings

Zero-shot text classification is often defined as training models on some seen labels and testing on an expanded set of both seen and unseen labels [Yogatama et al., 2017, Yin et al., 2019]. We use NatCat as a pretraining resource in this experimental setup.

We follow the same seen and unseen label splits as Yin et al. [2019], using their v0 and v1 splits, and same experimental setup. We train our models both starting from the original BERT-base-uncased model and the NatCat-pretrained BERT model. Table 5 summarizes the results (medians over 5 random seeds). The evaluation metrics are accuracy for Yahoo and label-weighted F1 for Emotion and Situation, in order to compare to Yin et al. [2019]. Pretraining on NatCat improves BERT's results on Yahoo, but it does not show clear improvements on Emotion and Situation in this setting.

Our Yahoo results are not directly comparable to the results from Yin et al. [2019] for several reasons, the most significant being that Yin et al. [2019] expand label names using their definitions in WordNet, while we choose to use the plain label names for our experiments. Also, Yin et al. [2019] formulate the problem as an entailment task, and there are differences in training set sizes. Yin et al. [2019] implement a "harsh policy" to impose an advantage to unseen labels by adding an $\alpha$ value to the probabilities of unseen labels. This $\alpha$ value is set by tuning on the development set which contains both seen and unseen labels. However, we do not assume access to a development set with unseen labels.

| | Topical (Acc.) | | | | | Sentiment (Acc.) | | | | | Multi label (LRAP) | | | | |
|---|---|---|---|---|---|---|---|---|---|---|---|---|---|---|---|
| | ag | dbp | yah | 20n | avg | emo | sst | yel | amz | avg | nyt | com | sit | avg | all |
| | | | | | | BERT models | | | | | | | | | |
| W | 72.3 | 86.0 | 49.0 | 33.3 | 60.5 | 21.3 | 63.8 | 64.5 | 67.0 | **66.6** | 41.8 | 24.3 | 51.1 | 39.0 | 53.0 |
| S | 69.0 | 76.0 | 59.1 | **51.2** | **64.0** | 18.7 | 60.1 | 57.8 | 57.0 | 59.1 | 36.5 | 24.1 | 49.9 | 36.8 | 51.0 |
| R | 70.3 | 72.8 | 51.8 | 49.2 | 61.7 | 12.5 | 61.2 | 67.0 | 66.2 | 65.2 | **49.8** | 22.6 | **52.4** | **41.5** | 52.1 |
| N | **75.6** | 82.8 | 54.9 | 39.3 | 63.3 | 16.1 | 62.7 | **70.4** | 63.6 | 53.8 | 49.6 | 22.6 | 50.5 | 41.0 | 53.3 |
| | | | | | | RoBERTa models | | | | | | | | | |
| W | 71.7 | **87.1** | 53.1 | 38.8 | 62.6 | **22.6** | 57.2 | 66.3 | **69.7** | 65.3 | 37.9 | 23.1 | 49.9 | 37.1 | 52.7 |
| S | 65.9 | 75.5 | **59.3** | 19.6 | 54.7 | 21.7 | 59.9 | 66.2 | 60.8 | 62.4 | 37.7 | **24.6** | 47.9 | 36.8 | 49.4 |
| R | 61.7 | 71.2 | 54.0 | 10.4 | 49.5 | 21.3 | 59.5 | 57.2 | 62.9 | 61.1 | 42.4 | 20.6 | 48.4 | 37.1 | 47.1 |
| N | 68.8 | 81.9 | 57.8 | 36.8 | 61.3 | 21.2 | **65.0** | 67.3 | 66.8 | 55.8 | 47.7 | 21.5 | 52.3 | 40.5 | **53.4** |

Table 6: Results of BERT and RoBERTa trained on individual NatCat data sources (W: Wiki., S: StackEx., R: Reddit, N: NatCat) and evaluated on CatEval.

The lower part of Table 5 shows the results of NatCat-trained models on the seen and unseen labels of these datasets. On Yahoo, a topical classification task, all models trained on the seen labels perform worse on the unseen labels (v0: 44.3%; v1: 34.9%) than RoBERTa trained purely on NatCat (v0: 52.1%; v1: 63.5%). Therefore, if we are primarily interested in unseen labels for topic classification tasks, our weakly supervised approach appears stronger than the more traditional half-seen setting. The NatCat-trained models also perform fairly well on unseen labels for Emotion and Situation, outperforming the unseen results from Yin et al. [2019] on the v1 split.

## 5. Analysis

### 5.1 Training Resources

Table 6 shows the model performances when trained on different resources of NatCat. For each single resource (Wikipedia, StackEx., or Reddit), we train the model for one epoch on 100k instances. We follow the exact same training procedure as described in Subsection 4.1.

Each data source is useful for some particular topical classification task, most of which can be explained by domain similarities. For example, models trained on Wikipedia are good at DBpedia, which can be explained by the fact that DBpedia is also built from Wikipedia. Stack Exchange is especially helpful for Yahoo; both are in the domain of community question answering. Models trained on Reddit, which contains a sizable amount of political commentary and news discussion in its most frequent categories, are particularly good at NYTimes. Models trained on Stack Exchange do not perform well on most sentiment related tasks. This is likely because Stack Exchange subareas are divided by topic. Wikipedia and Reddit are better resources for training sentiment classifiers, as they cover broader ranges of sentiment and emotion knowledge. It may be surprising that Wikipedia is good for sentiment classification. Wikipedia contains knowledge of sentiment words/phrases and their associations with positive or negative sentiment. For example, the Wikipedia category

of "emotion" https://en.wikipedia.org/wiki/Category:Emotion provides some examples of sentiment knowledge.

## 5.2 Training Set Size and Model Variance

While NatCat has over 10 million documents with over a million categories, we have used a small subset of it due to computational constraints. We compared models trained on 100k and 300k document-category pairs, following the same hyperparameter settings as in Section 4. We find that increasing training size generally harms performance on CatEval tasks. For example, averaging over all CatEval tasks, BERT trained on Wikipedia is 1.5 points lower when moving from 100k training instances to 300k instances. For Stack Exchange, the gap is 2.1 points. For Reddit, it is 0.7 points.

This is likely due to overfitting on the NatCat binary classification tasks. As there is a discrepancy between training and evaluation, increasing training data or epochs may not necessarily improve results on downstream tasks. This is a general phenomenon in weakly supervised and zero-shot classification, as we do not have development sets to tune training parameters for such tasks. Similar findings were reported by Puri and Catanzaro [2019], suggesting future work to figure out good ways to do model selection in zero-shot settings.

BERT and RoBERTa are known to suffer from instability in fine-tuning, i.e., training with different random seeds may yield models with vastly different results. We found both models have higher variance on sentiment tasks compared to topic classification. While nontrivial variances are observed (full results in Appendix E) ensembling the 5 models almost always outperforms the median of the individual models.

## 5.3 Error Analysis

Upon analysis of the confusion matrix of the RoBERTa ensemble predictions on AGNews, DBpedia, and Yahoo, we observe the following common misclassification instances:

- In AGNews, *science & technology* and *international* are often misclassified as *business*.
- In DBpedia, *nature* is often misclassified as *animal*, *nature* as *plant*, *written work* as *artist*, and *company* as *transportation*.
- In Yahoo, *society & culture* is often misclassified as *education & reference*, *politics & government*, and *business & finance*. *health* is often misclassified into *science & mathematics*, and *family & relationships* as *society & culture*.

The RoBERTa model trained on NatCat confuses closely related categories, but it rarely makes mistakes between clearly unrelated concepts. We find that human errors follow the same pattern: they are mostly in closely related categories. These results suggest that models trained on NatCat are effective at classifying documents into coarse-grained categories, but fine-grained categorization requires annotated training data specific to the task of interest.

## 6. Related Work

Wikipedia is widely used for weakly-supervised text classification. A classical work is the dataless text classification approach of Chang et al. [2008], Song and Roth [2014]. This approach uses Explicit Semantic Analysis [Gabrilovich and Markovitch, 2007], a method

to represent a document and a candidate category as sparse binary indicators of Wikipedia concepts and compute their relatedness by cosine similarity. Wang et al. [2009] learn a universal text classifier based on Wikipedia by extending the dataless approach. Yin et al. [2019] build models by directly mapping Wikipedia documents to their annotated categories, similar to the way that NatCat Wikipedia instances are created.

There has been a great deal of prior work on weakly-supervised and zero-shot text classification. For example, some have focused on restricted domains such as medical text [Mullenbach et al., 2018, Rios and Kavuluru, 2018], leveraged additional information such as semantic knowledge graphs [Zhang et al., 2019], or carefully exploited weak supervision such as class keywords [Meng et al., 2019] to achieve satisfactory performance. Some embed text and class labels into the same embedding space and use simple methods for classification [Dauphin et al., 2013, Nam et al., 2016, Li et al., 2016b, Ma et al., 2016]. Others model the presence of a special *unseen* class label and design specialized training and inference procedures to handle it [Shu et al., 2017, Fei and Liu, 2016, Zhang et al., 2019]. Yogatama et al. [2017] report zero-shot text classification experiments with a neural model that jointly embeds words and labels. The scale of NatCat is much larger than the small dataset-specific experiments of Yogatama et al. [2017], and the models trained with NatCat do not require additional supervision at test time such as seed words as in topic modeling approaches [Li et al., 2016a, Chen et al., 2015] or self-training [Meng et al., 2020]. Recent work [Zhang et al., 2021, Meng et al., 2020, Shen et al., 2021] uses naturally annotated hierarchical tree structure of label taxonomies to improve text classification. Puri and Catanzaro [2019] fine-tune language models on statements containing text and corresponding categories from Reddit. There is also a wealth of prior work in semi-supervised text classification: using unlabeled text to improve classification performance [Nigam et al., 2000, Howard and Ruder, 2018, Devlin et al., 2019, Liu et al., 2019, Lan et al., 2020, Peters et al., 2018].

## 7. Conclusion

We described NatCat, a resource of rich knowledge for text classification constructed from online community contributed content. We demonstrated that text classifiers built using NatCat perform strongly compared to analogous previous approaches. We have released the NatCat resource, CatEval benchmark dataset, our code for running experiments and for evaluation, and our best pretrained model at `https://github.com/ZeweiChu/NatCat`. The NatCat-trained models not only handle any label set but also supply a myriad of interpretable categories for a document off-the-shelf. We believe NatCat can be a useful resource for applications in natural language processing, information retrieval, and text mining.

## Acknowledgments

We wish to thank the anonymous reviewers for their feedback. This research was supported in part by a Bloomberg data science research grant to KS and KG.

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

| | | Topical (Acc) | | | Sentiment (Acc) | | | |
|---|---|---|---|---|---|---|---|---|
| | AG | DBP | YAH. | 20NG | Emo | SST | Yelp | Amz |
| GPT2 models without candidate answers | | | | | | | | |
| S | 55.8 | 43.7 | 31.1 | 25.1 | 14.1 | **66.7** | 66.4 | 70.0 |
| M | 56.4 | 35.3 | 32.7 | **28.1** | 17.3 | 66.2 | 69.5 | 71.8 |
| L | 51.1 | 42.6 | 36.7 | 21.8 | **17.7** | 60.4 | 65.8 | 69.5 |
| S+NC | 51.5 | 34.2 | 29.7 | 14.5 | 10.2 | 66.6 | 68.6 | 71.1 |
| M+NC | 49.9 | 42.2 | 28.2 | 13.0 | 11.5 | 53.2 | 61.5 | 58.6 |
| GPT2 models with candidate answers | | | | | | | | |
| M | 37 | 7.7 | 9.9 | 4.2 | 5.5 | 53.9 | 58.8 | 60.7 |
| M+NC | **72.5** | **72.6** | 28.4 | 4.2 | 14.9 | 57.7 | 60.2 | 63.7 |
| Puri and Catanzaro [2019] | | | | | | | | |
| 1/4 | 68.3 | 52.5 | **52.2** | - | - | 61.7 | 58.5 | 64.5 |
| All | 65.5 | 44.8 | 49.5 | - | - | 62.5 | **74.7** | **80.2** |

Table 7: GPT2 results. S/M/L are small/medium/large pretrained GPT2 models, and models with "+NC" fine-tune GPT2 on NATCAT.

## Appendix A. Preliminary Experiments with GPT2 Models

We report preliminary results in adapting GPT2 models to perform CATEVAL tasks. To do so, we construct the following descriptive text: "The document is about [category]: [document content]", where [category] is replaced by the class label we want to score, and [document content] is the document we want to classify. The descriptive text is tokenized by the BPE tokenizer, truncated to 256 tokens, and fed into the pretrained GPT2 model. The class label with the lowest average loss over all tokens is picked as the predicted label.

The results are shown in the initial rows of Table 7. We find mixed results across tasks, with the GPT2 models performing well on sentiment tasks but struggling on the topical tasks. Increasing GPT2 model size helps in some tasks but hurts in others.

We also fine-tune GPT2 models on NATCAT. Each document in NATCAT is paired with its category to construct the aforementioned descriptive text, and fine-tuned as a language modeling task.[11] The results (upper section of Table 7) are mixed, with the topical accuracies decreasing on average and the sentiment accuracies slightly increasing for GPT2 small but decreasing for GPT2 medium.

A key difference between training GPT2 and BERT/RoBERTa is that with GPT2, we do not explicitly feed information about negative categories. One way to incorporate this information is to construct descriptive text with "candidate categories" following Puri and Catanzaro [2019].[12] We sample 7 negative categories and 1 correct category to form the

---

11. The learning rate (set to 0.00002) follows linear warmup and decay. Following Puri and Catanzaro [2019], we set 1% of training steps as warmup period. We train for one epoch. The maximum sequence length is 256 tokens. We use batch size 8 for GPT2 small (117M parameters) and 2 for GPT2 medium (345M parameters).

12. The descriptive text is as follows: "`<|question|>` + question + candidate categories + `<|endoftext|>` + `<|text|>` + document + `<|endoftext|>` + `<|answer|>` + correct category + `<|endoftext|>`".

candidates. The results, shown in the middle section of Table 7, improve greatly for some tasks (AG, DBP, and Emo), but drop for the other CatEval tasks.

Puri and Catanzaro [2019] also create training data from Reddit. They annotate the text from each outbound weblink with the title of the Reddit post, and the subreddit that the link was posted in, while our Reddit dataset annotates each post title with the name of the subreddit it belongs to.

The GPT2 small model actually outperforms the 1/4-data training setting from Puri and Catanzaro [2019] on the sentiment tasks, though not the All-data training.

Compared to BERT and RoBERTa, it is harder to fine-tune a GPT2 model that performs well across CatEval tasks. In fact, there are many ways to convert text classification into language modeling tasks; we explored two and found dramatically different performance from them. It remains an open question how to best formulate text classification for pretrained language models, and how to fine-tune such models on datasets like NatCat.

## Appendix B. Hyperparameters and Model Training

The models and training hyperparametes are described in the main body of the paper. As we run our evaluations in a zero-shot setting, we do not have a development set for parameter tuning and model selection. All of our models (BERT, RoBERTa, GPT2) and training hyperparameters are chosen based on recommended settings from the Huggingface Transformers project [Wolf et al., 2019]. We perform 5 experiment runs with the same hyperparameter settings, except with different random seeds (1, 11, 21, 31, 41), and we report the median performances and standard deviations among different runs.

The following are the sizes of the models we use: BERT-base-uncased (110M), RoBERTa-base (110M), GPT-2 small (117M), GPT-2 medium (345M).

When performing supervised training on Emotion, NYTimes, Comment and Situation, all models are fine-tuned for 3 epochs following the same hyperparameter settings as the main experiment section. We train on only 1000 instances of NYTimes so we can finish training in a reasonable amount of time (less than 4 hours). Comment is trained on the test set as it does not have a training set, so its results represent oracle performance.

We perform our experiments on single GPUs (including NVIDIA 2080 Ti, Titan X, Titan V, TX Pascal). Training for a single epoch on 300k instances of NatCat takes about 450 minutes, and for 100k instances it will be around 150 minutes.

For evaluation on CatEval, it varies on different tasks. For small tasks like SST-2, it only takes about 20 minutes. For bigger tasks like Amazon-2, it takes about 1200 minutes on a single GPU for evaluation (we parallelize the evaluation over 40 GPUs so it only takes 30 minutes).

## Appendix C. Evaluation Metrics

We use three metrics to evaluate CatEval tasks.

**Accuracy.**    Accuracy is the number of correct predictions over the total number of predictions. It is used for single label classification tasks.

|  | NYT | COMM. | Situ. | AVG |
|---|---|---|---|---|
| BERT models | | | | |
| Wiki. 100k | 26.4 | 22.8 | 23.4 | 24.6 |
| Wiki. 300k | 25.2 | 22.1 | 25.7 | 24.3 |
| StackEx. 100k | 28.3 | 16.0 | 38.6 | 28.5 |
| StackEx. 300k | 27.9 | 13.1 | 37.8 | 26.5 |
| Reddit 100k | 36.4 | 13.0 | 32.5 | 27.7 |
| Reddit 300k | 35.4 | 11.7 | 28.9 | 25.4 |
| NATCAT | 32.5 | 16.1 | 37.1 | 28.4 |
| RoBERTa models | | | | |
| Wiki. 100k | 25.6 | 19.1 | 26.7 | 23.6 |
| Wiki. 300k | 24.4 | 18.0 | 29.8 | 24.0 |
| StackEx. 100k | 25.6 | 8.4 | 36.2 | 23.3 |
| StackEx. 300k | 23.5 | 7.6 | 32.7 | 21.0 |
| Reddit 100k | 36.4 | 8.5 | 34.6 | 24.2 |
| Reddit 300k | 35.4 | 5.5 | 29.4 | 21.7 |
| NATCAT | 31.0 | 13.4 | 36.2 | 27.2 |
| Other zero-shot | | | | |
| Yin et al. | - | - | 27.7 | - |

Table 8:   F1 scores of multi-label topic classification tasks

**Label Ranking Average Precision.**   This metric is used for evaluating multi-label text classification tasks, and is described in the main body of the paper. We use the scikit-learn implementation from `https://scikit-learn.org/stable/modules/generated/sklearn.metrics.label_ranking_average_precision_score.html`.

**Label Weighted F1 Score.**   This metric is used to evaluate multi-label text classification tasks. We follow the implementation from Yin et al. [2019].[13]

Table 8 compare the F1 scores of different models on multi label topic classification tasks.

## Appendix D. Training Sizes

Table 9 shows how the model performances vary with 100k or 300k training instances, following the same hyperparameter settings as in Section 4. We find that increasing training size generally harms performance on CATEVAL tasks. For example, averaging over all CATEVAL tasks, BERT trained on Wikipedia is 1.5 points lower when moving from 100k training instances to 300k instances. For Stack Exchange, the gap is 2.1 points. For Reddit, it is 0.7 points.

This is likely due to overfitting on the NATCAT binary classification tasks. As there is a discrepancy between training and evaluation, increasing training data or epochs may not necessarily improve results on downstream tasks. This is a general phenomenon in weakly supervised and zero-shot classification, as we do not have development sets to tune training

---

13. `https://github.com/yinwenpeng/BenchmarkingZeroShot/blob/master/src/preprocess_situation.py#L204`

|  | Topic | Senti. | Multi-label topic | All |
|---|---|---|---|---|
| BERT | | | | |
| Wiki. 100k | 60.5 | 55.3 | 39.0 | 53.0 |
| Wiki. 300k | 61.9 | 49.6 | 38.5 | 51.5 |
| StackEx. 100k | 64.0 | 49.0 | 36.8 | 51.0 |
| StackEx. 300k | 62.5 | 46.7 | 35.8 | 48.9 |
| Reddit 100k | 61.7 | 51.6 | 41.5 | 52.1 |
| Reddit 300k | 62.6 | 48.8 | 40.6 | 51.4 |
| NatCat 300k | 63.3 | 53.8 | 41.0 | 53.3 |
| RoBERTa | | | | |
| Wiki. 100k | 62.6 | 54.2 | 37.1 | 52.7 |
| Wiki. 300k | 62.7 | 55.4 | 36.9 | 52.8 |
| StackEx. 100k | 54.7 | 52.1 | 36.8 | 49.4 |
| StackEx. 300k | 52.8 | 51.9 | 35.4 | 47.4 |
| Reddit 100k | 49.5 | 51.1 | 37.1 | 47.1 |
| Reddit 300k | 47.9 | 51.8 | 37.4 | 46.2 |
| NatCat 300k | 61.3 | 55.8 | 40.5 | 53.4 |

Table 9: How training sizes affect model performances on CatEval

|  | Topic | Senti. | Multi-label topic | All |
|---|---|---|---|---|
| Wiki. | 1.1/0.6 | 3.1/2.8 | 0.5/0.3 | 1.3/1.2 |
| StackEx. | 0.8/1.2 | 0.7/2.6 | 0.5/0.7 | 0.2/1.2 |
| Reddit | 0.8/1.5 | 3.4/1.8 | 0.3/1.2 | 1.4/1.4 |
| NatCat | 0.8/1.2 | 3.6/1.8 | 0.7/0.4 | 1.3/0.2 |

Table 10: Standard deviations of BERT and RoBERTa model performances on CatEval tasks with 5 different random seeds.

parameters for such tasks. Similar findings were reported by Puri and Catanzaro [2019], suggesting future work to figure out good ways to do model selection in zero-shot settings.

## Appendix E. Model Variances

BERT and RoBERTa are known to suffer from instability in fine-tuning, i.e., training with different random seeds may yield models with vastly different results. To study this phenomenon in our setting, we performed training in each setting with 5 random seeds and calculate standard deviations for different tasks. As shown in table 10, both models have higher variance on sentiment tasks compared to topic classification. While nontrivial variances are observed, ensembling the 5 models almost always outperforms the median of the individual models.

## Appendix F. NatCat Category Names

We list the most frequent 20 categories from Wikipedia, Stack Exchange, and Reddit, separated by semicolons.

Wikipedia: years; births by decade; 20th century births; people; people by status; living people; stub categories; works by type and year; works by year; 20th century deaths; establishments by year; 19th century births; years in music; establishments by year and country; establishments by country and year; people by nationality and occupation; 1980s; alumni by university or college in the united states by state; 1970s; 1980s events

Stack Exchange: math; gis; physics; unix; stats; tex; codereview; english; gaming; apple; scifi; drupal; electronics; travel; ell; rpg; meta; mathematica; dba; magento

Reddit: AdviceAnimals; politics; worldnews; todayilearned; news; The_Donald; atheism; technology; funny; conspiracy; science; trees; gaming; india; soccer; WTF; reddit.com; Conservative; POLITIC; canada

## Appendix G. CatEval Category Names

We list all the category names all tasks in this section, separated by semicolons.

AGNews: international[14]; sports; business; science technology

DBpedia: company; educational institution; artist; athlete; politician; transportation; building; nature; village; animal; plant; album; film; written work

Yahoo: society culture; science mathematics; health; education reference; computers internet; sports; business finance; entertainment music; family relationships; politics government

20 News Groups: atheist christian atheism god islamic; graphics image gif animation tiff; windows dos microsoft ms driver drivers card printer; bus pc motherboard bios board computer dos; mac apple powerbook; window motif xterm sun windows; sale offer shipping forsale sell price brand obo; car ford auto toyota honda nissan bmw; bike motorcycle yamaha; baseball ball hitter; hockey wings espn; encryption key crypto algorithm security; circuit electronics radio signal battery; doctor medical disease medicine patient; space orbit moon earth sky solar; christian god christ church bible jesus; gun fbi guns weapon compound; israel arab jews jewish muslim; gay homosexual sexual; christian morality jesus god religion horus

Emotion: anger; disgust; fear; guilt; joy; love; no emotion; sadness; shame; surprise
SST-2: Negative; Positive
Yelp-2: Negative; Positive
Amazon-2: Negative; Positive

NYTimes: new england; real estate; news; britain; theater; new york and region; music theater and dance; your money; russia; iran; art and design; golf; candidates; campaign 2008; new york yankees; israel; pro basketball; healthcare; technology; media entertainment and publishing; family; manufacturing operations and logistics; banking finance and insurance; obituaries; california; media and advertising; health; travel; art; weddings and celebrations; legal; russia and the former soviet union; the city; asia; law enforcement and security;

---

14. The original name of this category in the dataset is "world". We chose "international" instead because "world" is not contained in ESA's vocabulary.

business; week in review; magazine; florida; plays; marketing advertising and pr; new jersey; international; long island; news and features; contributors; texas; style; west; education; sports; midwest; sunday travel; north america; asia pacific; science; book reviews; united states; westchester; editorials; middle east; markets; south; new york; china; addenda; medicine and health; europe; central and south america; movies; music; road trips; technology telecommunications and internet; washington d.c.; washington; baseball; new york city; arts; books; corrections; iraq; hockey; africa; japan; dance; government philanthropy and ngo; pro football; fashion and style; connecticut; germany; hospitality restaurant and travel; reviews; fashion beauty and fitness; food and wine; letters; usa; france; home and garden; americas; mid atlantic

COMMENT: team; player criticize; audience; sentiment; coach pos; team cav; player praise; team war; game expertise; game observation; refs pos; refs; stats; commercial; player humor; sentiment neg; injury; refs neg; feeling; sentiment pos; coach neg; player; commentary; play; coach; game praise; communication; teasing

SITUATION: water supply; search rescue; evacuation; medical assistance; utilities energy or sanitation; shelter; crime violence; regime change; food supply; terrorism; infrastructure; out of domain

