# OpenReview forum: "NATCAT: Weakly Supervised Text Classification with Naturally Annotated Resources"
_AKBC.ws/2021/Conference — AKBC 2021_

### Official Review · Reviewer_TZXz · 2021-07-18
**Useful resources + nice distantly supervised results; Old baseline + overfitted semi-supervised results**

**Rating:** 7
**Confidence:** 3

**Review:**

**Summary:** The paper considers the task of distantly supervised text classification. The model cannot access the labels in the target tasks. Instead, weak labels constructed from Wikipedia, Stack Exchange, and Reddit are used to train the model. The main contributions are:

- Scraped and formatted weakly labeled texts from Wikipedia (labels = categories), Stack Exchange (labels = subarea names), and Reddit (labels = subreddit names).
- A compiled benchmark with 11 datasets. The datasets are diverse (topic, sentiment, and emotion labels; single-label and multi-labels).
- A recipe to train the model on weak labels (apply BERT/RoBERTa on the input concatenated with the label, then predict a score). The paper also shows results on other potential recipes (e.g., cosine similarity to the label embedding, or generative models such as GPT-3) in the appendix.

**Strengths:**

1. The experiments are very thorough.
    - Alternative solutions considered are stated and their results are included in the appendix.
    - The experiments were on a diverse collection of tasks with different label spaces and numbers of labels per input.
    - The ablation and error analysis are also informative. The comparison against human raters also show that the types of errors made are reasonable.
2. The resources provided would be very useful for other researchers.

**Weaknesses:**

1. If I am not mistaken, the main baseline is quite old (a pre-neural model from 2008).
    - That said, there is a comparison against Yin18 at the end of Section 4.3. I wish this is used as the main baseline. Is these results comparable to the setup in Table 4?
2. In practical scenarios, a weakly supervised model is often fine-tuned on at least a small amount of target data (so that it's familiar with the target label space). The paper does consider this semi-supervised setup in Appendix D (the "Half Seen" setting). The result in Table 8 shows that the model greatly degrades on the "unseen" subset after fine-tuning on the "seen" subset, which is undesirable.
    - However, this overfitting issue might be fixable via regularization, such as a distance-based penalty when the parameters move too far from the initial parameters.
    - The overfitting problem seems to also manifest in Section 5.2, where training on a bigger training dataset hurts.

**Other comments:**

1. Section 3.1: "The choice of label names can have a large impact on performance." --- It would be interesting to do an ablation on the choice of target label names.
2. Section 4.1: Are the negative labels chosen randomly? Wouldn't such negatives be too easy to solve by the model?
3. Section 4.1: For the "encode the document and category using pretrained models" --- are these pretrained on the NatCat task? If not, how well does the cosine-similarity model tuned on NatCat perform?
4. Section 4.3: Why is the human performance on Yahoo much higher than the models (including the supervised one)?
5. It is still strange that the model trained on Wikipedia labels is so good at sentiment (Table 5). Are there Wikipedia categories that resemble sentiment or emotion labels? The Reddit results is more understandable due to sentiment-like subreddit names (like oddlysatisfying or mildlyinfuriating).

**Edit suggestions:**
- Section 1: "StackOverflow" --> "Stack Exchange"?
- Section 4.3: Yin 18 has an incorrect citation. Should be "Benchmarking Zero-shot Text Classification: Datasets, Evaluation and Entailment Approach"
- Section 6: "Yin et al. [2018] and" --- wrong Yin18 and a missing second reference.

---

> ### Author Response · Authors · 2021-07-30
> **Clarifications on some experimental results**
>
> > If I am not mistaken, the main baseline is quite old (a pre-neural model from 2008).
> That said, there is a comparison against Yin18 at the end of Section 4.3. I wish this is used as the main baseline. Is these results comparable to the setup in Table 4?
>
> These results are comparable. We will expand on those results in the next version by clarifying that they are comparable and discussing the comparison in more detail.
>
> > Section 3.1: "The choice of label names can have a large impact on performance." --- It would be interesting to do an ablation on the choice of target label names.
>
> Thanks for the advice! We can do that and add some results in the main body (if space permits) or appendix.
>
> > Section 4.1: Are the negative labels chosen randomly? Wouldn't such negatives be too easy to solve by the model?
>
> Yes, they are randomly sampled. We did not sample harder negative examples in our experiments.
>
> > Section 4.1: For the "encode the document and category using pretrained models" --- are these pretrained on the NatCat task? If not, how well does the cosine-similarity model tuned on NatCat perform?
>
> These models are not fine-tuned on the NatCat resource. With the model setup described in section 4.1 and cosine similarity as the scoring function, we get model prediction accuracies of 30.2% for AGNews, 30.0% for DBPedia, and 19.0% for Yahoo. Compared to 75.6% for AGNews, 82.8% for DBPedia, and 54.9% for Yahoo from Table 4
>
> > Section 4.3: Why is the human performance on Yahoo much higher than the models (including the supervised one)?
>
> Good observation/question. The Supervised result for Yahoo is cited from Zhang et al.’s CharCNN text classification paper (https://arxiv.org/pdf/1509.01626.pdf). With BERT we could expect higher accuracy (77.6% in Sun et al. https://arxiv.org/pdf/1905.05583v3.pdf ). Hence Yahoo is not an outlier in this sense, but it is indeed more noisy and harder to solve. As discussed in the paper: “In Yahoo, society & culture is often misclassified as education & reference, politics & government, and business & finance. Health is often misclassified into science & mathematics, family relationships as society culture.” These categories are hard to tell and can be classified in multiple categories. We can update the final version with more recent BERT based supervised results.
>
> > It is still strange that the model trained on Wikipedia labels is so good at sentiment (Table 5). Are there Wikipedia categories that resemble sentiment or emotion labels? The Reddit results is more understandable due to sentiment-like subreddit names (like oddlysatisfying or mildlyinfuriating).
>
> We think Wikipedia is likely to contain some knowledge about sentiment words and phrases, such as certain words or phrases belonging to certain emotion, and that emotion can be associated with positive or negative sentiment. The category of emotion (https://en.wikipedia.org/wiki/Category:Emotion) may provide a lot of such knowledge.

---

### Official Review · Reviewer_MExX · 2021-07-20
**Good general-purpose text classification resource with lack of proper validation**

**Rating:** 6
**Confidence:** 3

**Review:**

# General comments
This paper presents a dataset and evaluation benchmark for weakly supervised text classification. The main resource, NatCat, contains a large number of diverse real-world documents and categories, making it appealing for general-purpose off-the-shelf text classification. The way of generating weakly supervised data is efficient, intuitive and easy to reproduce. The baseline models trained on this resource easily outperform previous weakly supervised approaches.

The main weakness of this work stems from the lack of support for its main hypothesis. This is a training and evaluation resource but instead of comparing against other resources, the main comparison (Tables 4, 5) is between multiple models trained on the same resource. If the motivation for this work is that NatCat is a better resource for text classification, the experiments need to demonstrate exactly that.

Similarly for the CatEval benchmark - currently, the contribution is the consolidation of multiple existing evaluation datasets. There is nothing in the paper to test if CatEval is a better resource than existing benchmarks for this task or whether the datasets comprising CatEval are the best choices for the purpose.

Another practical consideration regarding any weakly supervised †raining resource is how much domain-specific fully supervised data does it take to get a better performance.


# Specific comments/questions
It's not clear what the example is Table 1 illustrates. Are the highest-scoring categories from NatCat a good fit? As there is no overlap with the AGNews categories (including the true one) wouldn't the score of the example be 0? Furthermore, if the categories from AGNews are part of NatCat (as is claimed in Section 3.1) why are they not appearing even further down the list?

None of the datasets used in CatEval has nearly as many labels as there are in NatCat (especially the Wikipedia part). Does this cause problems for the models? Would it be better for the number of categories in NatCat to be consolidated to match those found in CatEval?

Why are only the common categories (subreddits) chosen from Reddit, but not Wikipedia? What are some examples of subreddits that fell below the threshold?

The result of "superhuman" performance is a bit suspicious. To confirm this is semantic drift from the annotators' side, the same annotation should be performed with training sessions for each annotator. The alternative possibility is that the categories aren't applied consistently among the users of the websites comprising the dataset (which is plausible in community-generated content).

I disagree with the analysis in 5.1 claiming that Wikipedia is a better resource for training sentiment classifiers than Stack Exchange. Wikipedia's editorial style aims to be as objective as possible.


# Minor comments
- Incorrect capitalisation in the second paragraph of Section 1 ("of NatCat, We" -> "of NatCat, we")
- Inconsistent non-textual citation style applied to some references ([Yogatama et al., 2017] instead of Yogatama et al. [2017])
- Add the #words/doc statistic to Table 2.
- I don't think median was the statistic used in reporting the results over 5 runs (Section 4.1). Maybe the authors meant mean? Also, the standard deviation/error should be reported for completeness.

---

> ### Author Response · Authors · 2021-07-30
> **Clarification of the NatCat resource and some experimental settings**
>
> Thanks for the valuable feedback! Below are our responses.
>
> > It's not clear what the example is Table 1 illustrates ...
>
> Table 1 is an example to show that NatCat can be used to score any document-category pairs for text classification tasks. The row “NatCat” shows categories from the NatCat resource. In particular, these categories are from the Wikipedia dataset. We put this row here to show that models trained on NatCat can be used to classify common categories (from Wikipedia). However, if we want to use NatCat for a particular text classification task, we may also just treat it as a scoring and ranking task, to first score document-category pairs of each category, and pick the highest scored category as the predicted label. Thanks for pointing this out, we will add more descriptions in the caption of Table 1, so the readers won’t be confused.
>
> AGNews categories do appear further down the list. The list is so long (millions of labels) and the AGNews labels are not as high scoring as the labels shown. The NatCat labels as shown in Table 1 are indeed closer to the news text compared to “International” (Label from AGNews).
>
> > None of the datasets used in CatEval has nearly as many labels as there are in NatCat (especially the Wikipedia part). Does this cause problems for the models? Would it be better for the number of categories in NatCat to be consolidated to match those found in CatEval?
>
> We don’t think this is necessary. Our way of using NatCat to fine-tune BERT/RoBERTa models essentially builds models that are capable of scoring any document-category pairs. Presumably, it learns to capture the semantic relatedness or similarity between the document text and category name or descriptions. Our goal is to make such a model as general-purpose as possible and can be applied to a variety of downstream text classification tasks. With the NatCat resource constructed and made available, future work could explore several interesting follow-up questions, including whether it's beneficial to focus training on a small number of high-level categories.
>
> > Why are only the common categories (subreddits) chosen from Reddit, but not Wikipedia? examples of subreddits ...
>
> We find subreddit names are generally more noisy, and a lot of them are abbreviations that only a certain group of people knows. Wikipedia category names are more formal and understandable by most people. Hence we decide to only keep the most common categories from Reddit. Below are some example category names from Subreddit (with its ranking by frequency): archeage (3001), Deutsche_Queers (10000), HartfordWhalers (20000), pyladies (30000), Religitards (40000), TheArnoldFanClub (50000), HomeSeer (60000), 001 (68627).
>
> > The result of "superhuman" performance is a bit suspicious ...
>
> That’s a good point. We actually want to convey the message that “text classification” can suffer from overfitting, in a sense that if human beings cannot outperform machine-trained models, that “superhuman” performance may not be that meaningful. In this sense, NatCat fine-tuned models do not seek to outperform or even get close to state-of-the-art models on any particular text classification tasks, because a real person with reasonable world knowledge won’t outperform state-of-the-art machine-learned models. As you were getting at, we suspect the reason why human performance is below that of supervised models is that the categories were generated naturally by online communities rather than by annotators with annotation guidelines or training examples. We chose to conduct our human evaluation with the setting in which humans are solely given label names rather than guidelines or training examples, in order to reflect the setting of our NatCat-trained classifiers.
>
> > I disagree with the analysis in 5.1 claiming that Wikipedia is a better resource for training sentiment classifiers than Stack Exchange. Wikipedia's editorial style aims to be as objective as possible.
>
> Yeah, that’s a good point. We were interpreting the results from Table 5 directly here. Wikipedia is likely to contain more knowledge about sentiment words and phrases, such as certain words or phrases belonging to a certain emotion, and that emotion can be associated with positive or negative sentiment. StackExchange categories are very engineering-focused and may provide less knowledge for sentiment classification tasks. However, Table 5 shows that none of Wikipedia, StackExchange, or Reddit is a perfect resource for sentiment classification tasks. We can soften the tone in the final version of the paper to say that Wikipedia provides more knowledge about sentiment in its category structure compared to Stack Exchange, but it is by relative measure. In general, NatCat is not ideal for sentiment tasks but good for topical classification.

---

> > ### Comment · Reviewer_MExX · 2021-07-30
> > **Thanks for the detailed clarifications**
> >
> > I would like to thank the authors for providing clear and detailed answers to my questions. Regarding subreddit naming your point makes a lot of sense and the examples are helpful to understand the reason of the threshold of 3k. However, I suspect there are subreddit names above the threshold that have the same problem of "in-group meaning" - (e.g. r/Superstonk or r/facepalm). A more in-depth analysis of the categories that are in the dataset would be interesting.
> >
> > I would also like to encourage you to have a look at my general comments, especially the point about providing evidence about the main hypothesis put forward by this work. While NatCat is undeniably a useful resource, I believe you can make a much stronger researcher paper if you address these issues.

---

> > > ### Author Response · Authors · 2021-07-30
> > > **Thanks for the advice! Add more response.**
> > >
> > > Thanks for the advice! Here are some more response.
> > >
> > > > A more in-depth analysis of the categories that are in the dataset would be interesting.
> > >
> > > We can work on that and add some analysis of category names in the paper as you suggested.
> > >
> > > > I would also like to encourage you to have a look at my general comments, especially the point about providing evidence about the main hypothesis put forward by this work. While NatCat is undeniably a useful resource, I believe you can make a much stronger researcher paper if you address these issues.
> > >
> > > We could compare with Yin et al. However, Yin’s work is also based on Wikipedia (document and their immediate categories). This is similar to what we demonstrated in Table 5, i.e., with resource Wikipedia only. We could add some text in the paper saying that “WikiOnly” is close to Yin et el., except we did more processing of category names and parent categories, as described in “Section 2 → Wikipedia”.
> > >
> > > We could also provide a comparison to other pretrained language models, as the other reviewer suggested. Such language models are trained purely on text without naturally annotated categories. When classifying with the pretrained BERT model (not fine-tuned on NatCat), following the model setup described in section 4.1 with cosine similarity as the scoring function, we get model prediction accuracies of 30.2% for AGNews, 30.0% for DBPedia, and 19.0% for Yahoo. Compared to 75.6% for AGNews, 82.8% for DBPedia, and 54.9% for Yahoo from Table 4, fine-tuning on NatCat greatly improves model performance on text classification tasks. We could add such results as another baseline in the paper.
> > >
> > > > Add the #words/doc statistic to Table 2.
> > >
> > > Yes, we can add them.
> > >
> > > > I don't think median was the statistic used in reporting the results over 5 runs (Section 4.1). Maybe the authors meant mean? Also, the standard deviation/error should be reported for completeness.
> > >
> > > We actually meant median. This was following the approach of the RoBERTa paper (https://arxiv.org/pdf/1907.11692.pdf). The reason is that sometimes RoBERTa fine-tuning could fail and give very bad results. Training on 5 random seeds and report the median provides more robust results.

---

### Official Review · Reviewer_Bz1V · 2021-07-22
**Interesting new source of weak supervision. Would be stronger with comparison to more recent work in zero-shot classification.**

**Rating:** 6
**Confidence:** 4

**Review:**

This paper presents a method of extracting document class labels from the Wikipedia category hierarchy; the set of sub-areas on StackOverflow; and the list of sub-reddits on Reddit. These pairs are then used to train a BERT-based binary classifier that predicts whether a category matches a document. Since categories are named with natural language, the model can be applied to new classification tasks that have natural language labels.

When applied to a number of few-shot classification taskss, the model outperforms baselines that don't make use of large pre-trained language models. On average, it is beneficial to use all three sources of distant supervision, but this is not true for each individual task.

The results would be stronger with a comparison to a zero-shot approach that makes use of a large pre-trained LM but doesn't make use of the source of weak supervision. Currently, it is not clear where the accuracy improvements are coming from. An alterative way to show the importance of the NatCat data would be to report accuracy results for the BERT/RoBERTA models that have been trained on 1%/10%/50%/100% of the available dataset.

Strengths:
- Aggregation of three different sources of weak supervision is yielding nice results on zero-shot classification.
- Data may be useful resource for future work.

Weaknesses:
- Experiments don't properly untangle impact of NatCat weak supervision vs. adoption of large pre-trained LM.
- No comparison to recent work on zero-shot classification.

---

> ### Author Response · Authors · 2021-07-30
> **Performance of Pretrained LM without NatCat and Comparison to recent zero-shot classification.**
>
> Thanks for the feedback! Below are our responses:
> > Experiments don't properly untangle impact of NatCat weak supervision vs. adoption of large pre-trained LM.
>
> When classifying with the pretrained BERT model (not fine-tuned on NatCat), following the model setup described in section 4.1 with cosine similarity as the scoring function, we get model prediction accuracies of 30.2% for AGNews, 30.0% for DBPedia, and 19.0% for Yahoo. Compared to 75.6% for AGNews, 82.8% for DBPedia, and 54.9% for Yahoo from Table 4, fine-tuning on NatCat greatly improves model performance on text classification tasks.
>
> > No comparison to recent work on zero-shot classification.
>
> We compared to Yin et al. at Table 8 in the appendix. We also compared to Puri and Catanzaro (GPT2) at Table 6 in the appendix. We could move that up to the main text in the final version. These were the zero-shot approaches we were aware of. Are there more works reviewers would suggest us to compare to?

---

### Decision · Program_Chairs · 2021-08-17

**Decision:**

Accept

**Comment:**

The reviewers agree that the resource presented is likely to be of interest to the community. However, they also point out that the paper has methodological issues. e.g. the resource is not compared as such to any other resources to demonstrated its value. The models used in the comparisons are out of date, and some of the conclusions drawn are over-stated.  We strongly urge the authors to incorporate reviewers' comments and address these issues in the camera-ready version.